# Investigation of the Appropriate Reverberation Time for Lower-Grade Elementary School Classrooms Using Speech Intelligibility Tests

**A-Hyeon Jo [1]** , **Chan-Jae Park [2]** **and Chan-Hoon Haan [1],***

1 Department of Architectural Engineering, Chungbuk National University, Cheongju 28644, Korea; ahyunlove04@naver.com
2 Korea Institute for Local Educational Finance, Sejong-si 30130, Korea; cjpark@cbnu.ac.kr
* Correspondence: chhaan@cbnu.ac.kr

**Abstract:** Because speech recognition performance is significantly lower at the age of nine or younger, the acoustic performance standards of classrooms for young children should be investigated. This study derives the appropriate reverberation time for lower-grade elementary school classrooms in Korea. A virtual sound field was created by computer modeling using normalized classrooms of Korean elementary schools. A total of five classrooms with reverberation times from 0.6 s to 1.2 s were produced by varying the sound absorption coefficient of the ceiling. Auralized sounds were produced by synthesizing anechoic sounds of words in a virtual sound field. Speech intelligibility tests were undertaken with 20 young students, aged nine. As a result, it was concluded that the reverberation time suitable for lower-grade classrooms of elementary schools should be below 0.6 s because test scores were significantly lower when RT was longer than this value.

**Keywords:** classroom; lower-grade elementary school students; reverberation time; speech intelligibility test

## 1. Introduction

School is an educational institution, and the classroom is a space where students receive an education. Education at school is received primarily through teachers' voices, and hence, speech transmission is critical in classrooms. Sound intelligibility is a very important component with which to determine classroom acoustic performance. Of a variety of acoustic factors, reverberation time and background noise are crucial in influencing acoustic performance. Many studies have concluded that reverberation time is strongly associated with academic achievement. In classrooms with a short reverberation time, the speech intelligibility test score was high even in a noisy environment [1]. The reverberation time was found to greatly influence listening effort (LE) [2], and a 50 ms early/late sound ratio, the reverberation time, and the background noise level were identified as factors influencing speech intelligibility in classrooms [3]. To provide students with an acoustic environment adequate for learning, classroom acoustics standards should be available.

Currently, the United States and United Kingdom use acoustics standards for reverberation time and background noise in classrooms. Table 1 shows U.S. standards according to classroom volume and type based on the acoustics criteria stated in the ASA S12.60 [4], and Table 2 shows U.K. standards according to student age and classroom use as stated in Building Bulletin 93 [5].

In most countries, the classroom background noise standard is around 30~35 dBA. However, the standard for reverberation time varies from country to country. The reverberation time standard is 0.6 s in Chile, Norway, and Denmark, and 0.8 s in Germany, France, Italy, China, and Malaysia. In some other countries, the standard is set between 0.4 s and

1.2 s depending on classroom size and student age. In contrast, South Africa and India do not have standards [6–9].

**Table 1.** Acoustic performance standards for classrooms in the United States (unoccupied). m$^3$.

| Classroom Size (Volume) | RT (s) | BN (dBA) |
|---|---|---|
| less than 283 m$^3$ | <0.6 | 35 |
| 283 m$^3$~566 m$^3$ | <0.7 | 35 |
| more than 566 m$^3$ | no requirement | 40 |

**Table 2.** Acoustic performance standards for classrooms in the United Kingdom (unoccupied).

| Classroom Type | | RT (s) | BN (dBA) |
|---|---|---|---|
| elementary school | | <0.6 | <35 |
| middle school | | <0.8 | <35 |
| open-plan classroom | | <0.8 | <40 |
| lecture room | less than 50 people | <0.8 | <35 |
| | more than 50 people | <1.0 | <30 |

In Korea, various studies have been conducted on classroom acoustic environments. Some studies investigated the acoustic performance of school classrooms built between 1984 and the present using a variety of construction methods. The findings indicated that the average reverberation time was under 0.8 s, whereas the reverberation time in old classrooms only was over 1.0 s [10,11]. A study investigated the impact of an uneven distribution of sound absorption inside the classroom on inter-aural level difference and speech intelligibility in students. As the sound absorption coefficient of the side wall increases, the sound level difference between the two ears increases, and speech intelligibility was decreased. According to these results, there is less effect on the sound level difference between two ears by adjusting the sound absorption coefficient of the ceiling material rather than the side walls of the classroom [12]. The correct answer rates of a speech intelligibility test were different depending on the language used, such as English, Chinese, or Korean, in the same acoustic conditions. Furthermore, the correlation between the reverberation time and speech intelligibility was also different for various languages. It was shown that factors affecting speech intelligibility differed depending on the pronunciation characteristics of the languages. Therefore, the acoustic standards for school classrooms should be set differently depending on the language used [13].

Currently, Korea does not have classroom acoustics standards, except for noise, which should be 55 dBA or lower according to the School Health Act [14]. Hence, studies were undertaken to develop recommendations for acoustics standards for school classrooms in Korea. Speech intelligibility tests were performed in a variety of reverberation environments to recommend reverberation time and background noise standards appropriate for middle and high school classrooms, as shown in Table 3 [15], and interior finishing guidelines were proposed to meet the standards [16]. The proposed standards are specifically for classrooms of middle and high school students (age 14 or older).

**Table 3.** Suggested acoustic performance standards for classrooms in Korea (unoccupied).

| Target | RT (s) | BN (dBA) |
|---|---|---|
| middle and high school (below 220 m$^3$) | <0.8 | <35 |

In Korea, the age range for children from elementary to high school is 6–18. Students' speech perception ability changes with age. According to a study that utilized the Hearing in Noise Test (HINT), from age 14, speech perception abilities in children start to be similar

to those of adults; in a noisy environment, children's ability decreased as age decreased, and it was markedly lower under age 10 [17]. Accordingly, classroom acoustics standards should vary according to student age by taking age differences in auditory cognitive ability into account.

Researchers in countries other than Korea have also recognized the need to differentiate classroom acoustics standards according to age. In Santiago, Chile, an issue was raised with the existing reverberation time standards for kindergarten and elementary and middle school classrooms because they were identical regardless of student age [6]. A study summarized the recommendations regarding classroom acoustic performance in different countries and proposed age-specific guidelines for background noise and reverberation time for age groups of 6~7, 8~9, and 10~11 [18].

The study in which the aforementioned classroom acoustics standards in Korea were developed was conducted with students of age 14 or higher. Hence, because elementary school students are not fully developed in their speech perception abilities, the standards may be inappropriate for elementary school classrooms. Additionally, considering speech perception ability is lower in children aged 10 or younger, standards specific to lower-grade elementary school classrooms should be developed.

Accordingly, the major goal of the present study is to suggest an appropriate reverberation time for Korean lower-grade elementary school classrooms.

For this purpose, the standard model of the Korean lower-grade elementary school classroom was reproduced, and various reverberation conditions were created by changing the sound absorption using a computer simulation. Additionally, auralized sounds with various reverberation times were created. Using these auralized sounds, speech intelligibility tests were performed with lower-grade elementary school students and adults, and a reverberation time appropriate in lower-grade elementary school classrooms was derived.

## 2. Previous Studies

A previous study was conducted to investigate the current status of lower-grade elementary school classrooms in Korea. Specifically, physical parameters of acoustic performance were measured in lower-grade classrooms of elementary schools in Cheongju to examine the classroom acoustic environment (such as reverberation time and background noise) [19].

A total of three classrooms of lower-grade students were selected in elementary schools located in Cheongju, and physical parameters of classroom acoustic performance were measured. The data of three additional classrooms that have already been measured were added to the analysis, and the acoustic performance of a total of six classrooms was examined. Table 4 presents the information of the six classrooms, such as construction year and architectural dimension.

**Table 4.** Information about the classification and grade of classrooms.

| Classification | Construction Year | Architectural Dimension of the Classroom | | | | | Grade of Classroom |
|---|---|---|---|---|---|---|---|
| | | Length (L) | Width (W) | Height (H) | Volume (V) | Ratio (L:W) | |
| N | 1999 | 9.4 m | 7.8 m | 2.8 | 206.6 | 1:0.834 | 2 |
| K1 | 1983 | 8.3 m | 7.5 m | 2.6 | 160.0 | 1:0.904 | English classroom |
| K2 | 1983 | 8.8 m | 7.0 m | 2.9 | 177.9 | 1:0.793 | 1 |
| C | 1981 | 8.8 m | 7.0 m | 2.9 | 178.6 | 1:0.795 | 2 |
| S | 2007 | 8.4 m | 7.4 m | 2.6 | 161.6 | 1:0.881 | 2 |
| H | 2008 | 8.2 m | 7.7 m | 2.6 | 164.2 | 1:0.939 | 1 |

To avoid the impact of external noises during measurement, all doors and windows were shut, and measurement was performed based on KS F 2864 [20]. To minimize the effects of variables like clothing and noise from people inside, the classrooms were unoccupied during measurement. In the experiment, a directional speaker was used as a

sound source, matching the directivity pattern of the teacher who speaks to the students during class. The sound source was positioned in the center of the podium in front of the blackboard in consideration of the teacher's location and at a height of 1.5 m from the floor, considered the average location of the mouth of an adult. The locations of the sound-receiving points were set up differently when measuring background noise and reverberation time. The background noise levels were measured at the center of the classroom and near the window. Also, reverberation times were measured at 9 points that were evenly distributed in the classroom. Sound receivers (microphones) were located at a height of 1 m from the floor to correspond with the location of the ears of lower-grade elementary school students sitting on chairs.

The locations of the sound source and sound-receiving points are presented in Figure 1. Measured background noise and reverberation time in the classrooms are illustrated in Figure 2.

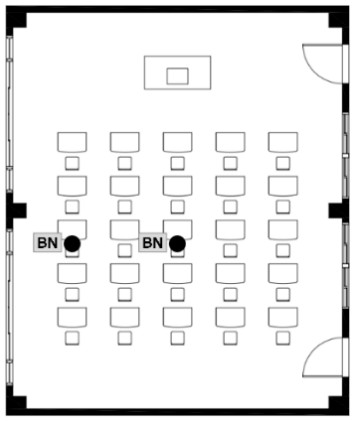

(**a**) Background noise level

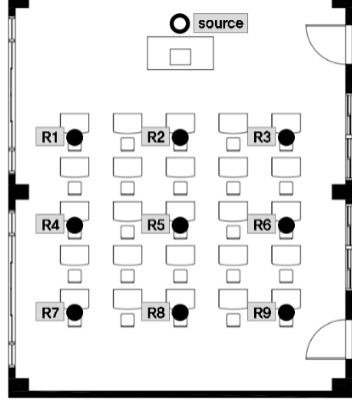

(**b**) Reverberation time, $T_{30}$

**Figure 1.** Location of sound source and receiving points for measurement in the elementary classroom.

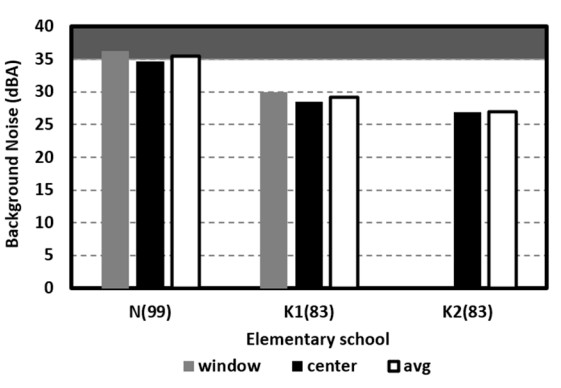

(**a**) Background noise

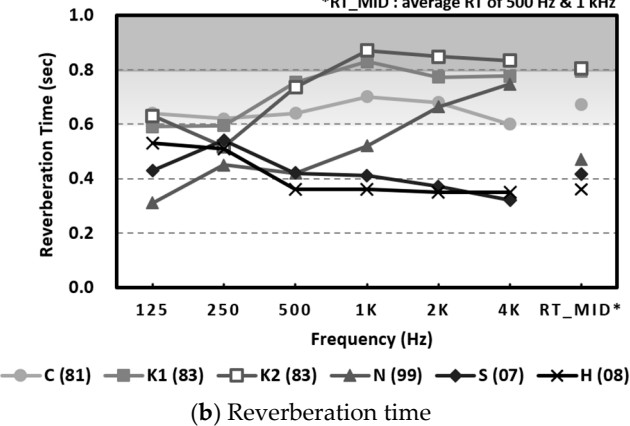

(**b**) Reverberation time

**Figure 2.** Measurement results of background noise levels and reverberation time in Korean elementary school classrooms (unoccupied).

In Figure 2a, the gray bars represent the measurements near classroom windows, and the black bars indicate the measurements in the centers of the classrooms. The background noise level of N elementary school exceeded the US, UK and Korean standards of 35 dBA. In all the classrooms, background noise near windows was more intense.

The measured reverberation times in each classroom are presented in Figure 2b. As shown in Figure 2b, $RT_{mid}$ did not exceed 0.8 s in all six classrooms. However, in three of the classrooms, $RT_{mid}$ was longer than 0.6 s, which is the standards of reverberation time

in US and UK. It is inferred that this is because three schools were built relatively recently and the ceilings were finished with sound-absorbing boards efficient for high-frequency sound, whereas the ceilings in schools built in the 1980s were finished with paint over sound-absorbing boards which eventually reduced the sound absorption performance.

According to previous studies, classrooms built in different years has different reverberation times. Some classrooms met the Korean middle and high school classroom acoustics standards but, it was not confirmed that these classrooms can provide an adequate listening environment for lower-grade elementary school students. In conclusion, acoustic standards specific for them should be developed to provide appropriate listening environment for lower-grade elementary school students in Korea.

## 3. Research Method

This study seeks to derive a reverberation time ($RT_{mid}$) appropriate in lower-grade elementary school classrooms based on a speech intelligibility test. To do so, a model of a standard lower-grade elementary school classroom was reproduced, and virtual sound fields with different reverberation times were created by using various finishing materials. A total of five reverberation times (0.4 s, 0.6 s, 0.8 s, 1.0 s, and 1.2 s) were used, and auralized sounds for testing were produced in each of the five virtual sound fields. The speech intelligibility tests used the auralized sounds created for each value of reverberation time. The test subjects were lower-grade elementary school students whose hearing was underdeveloped and adults with normal hearing.

### 3.1. Creation of a Virtual Sound Field in the Standardized Classroom

To create a virtual sound field, a 3D model of a standard classroom for lower-grade elementary school students was created using AutoCAD. Korean classrooms have been standardized under the regulation of government, which means that almost all general Korean elementary school classrooms are rectangular with flat ceilings. The size of the classroom was determined based on the average interior dimensions of classrooms in 10 elementary schools in Cheongju, that is, 7.3 m in length, 8.5 m in width, and 2.6 m in height. The information of the 10 elementary school classrooms is reported in Table 5 [21]. The size, floor plan with desk arrangement, and cross-sectional layout of the standard classroom are presented in Figure 3, and its 3D model is displayed in Figure 4.

**Table 5.** Information of the 10 elementary school classrooms.

| Construction Year | District Classification | Architectural Dimension of the Classroom | | | | |
| --- | --- | --- | --- | --- | --- | --- |
| | | Length (L) | Width (W) | Height (H) | Volume (V) | Ratio (L:W) |
| 1971 | general area | 7.3 m | 8.8 m | 2.4 m | 154.2 m$^3$ | 1:1.21 |
| 1972 | roadside area | 7.9 m | 8.0 m | 2.6 m | 164.3 m$^3$ | 1:1.01 |
| 1974 | general area | 7.4 m | 8.8 m | 2.8 m | 182.3 m$^3$ | 1:1.19 |
| 1981 | general area | 7.0 m | 9.0 m | 2.9 m | 178.6 m$^3$ | 1:1.26 |
| 1987 | general area | 7.0 m | 9.0 m | 2.7 m | 175.8 m$^3$ | 1:1.19 |
| 2003 | roadside area | 7.4 m | 8.0 m | 2.6 m | 153.9 m$^3$ | 1:1.08 |
| 2007 | roadside area | 7.2 m | 8.2 m | 2.5 m | 147.6 m$^3$ | 1:1.14 |
| 2007 | general area | 7.4 m | 8.4 m | 2.6 m | 161.6 m$^3$ | 1:1.14 |
| 2008 | general area | 7.2 m | 8.2 m | 2.6 m | 153.5 m$^3$ | 1:1.14 |
| 2008 | roadside area | 7.7 m | 8.2 m | 2.6 m | 164.2 m$^3$ | 1:1.06 |
| average | | 7.3 m | 8.5 m | 2.6 m | 163.3 m$^3$ | 1:1.14 |

Finishing materials typically used in each area of an elementary school classroom were inputted in the 3D classroom model using the room acoustic simulation software Odeon [22]. Because reverberation time is influenced by classroom volume and the sound absorption rates of finishing materials, the reverberation time in each experimental condition was adjusted based on the sound absorption rates of such materials. In consideration of the fact

that the standard for middle and high school classrooms in Korea is 0.8 s, a total of five reverberation times, 0.4 s, 0.6 s, 0.8 s, 1.0 s, and 1.2 s, were set as experimental conditions.

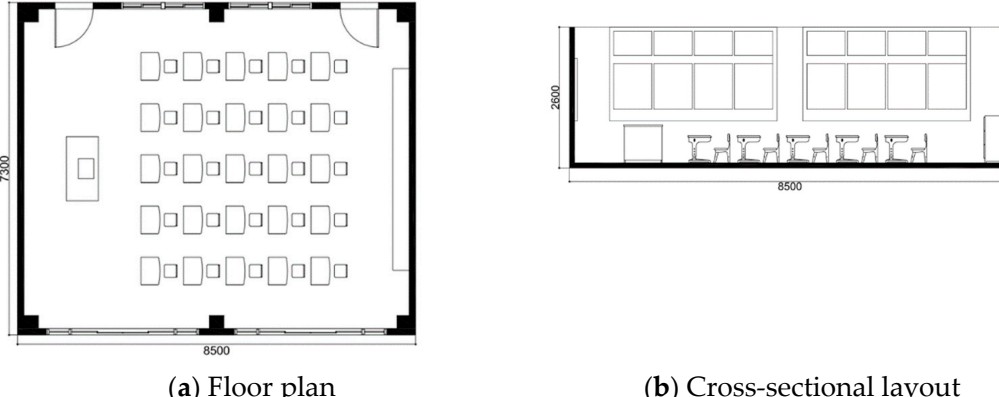

(**a**) Floor plan          (**b**) Cross-sectional layout

**Figure 3.** Floor plan and cross-sectional layout of standard elementary school classroom in Korea (unit: mm).

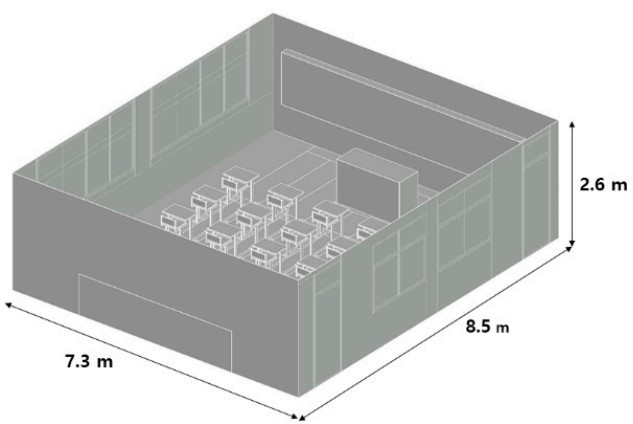

**Figure 4.** 3D model of standard elementary school classroom in Korea.

As the sound absorption coefficient of the side wall increases, the sound level difference between the two ears increases, and speech intelligibility is decreased [12]. Since the ceiling is the surface which reflects the teacher's voice first, the clarity and reverberation of the classroom can be effectively adjusted when the ceiling is treated as absorptive or reflective. Therefore, the reverberation time was adjusted by manipulating the sound absorption coefficient of the finishing materials of the ceiling rather than the side walls.

The types and sound absorption coefficients of interior finishing materials by location are presented in Table 6.

NC (noise criteria) values were set to deal with background noise in the classroom during the simulation. The NC value refers to the level of noise that can be allowed indoors and was set to NC 30 in consideration of the level of noise allowed in the classroom [23]. The same background noise was set for all reverberation times.

The sound source and receiving points were set to follow the current teaching methods during the simulation. The sound source was located at the center of the podium in front of the blackboard (a typical place where teachers teach the class) and at a height of 1.5 m from the floor (the average location of the mouth of a male adult). The location of the sound-receiving points was at a height of 1.2 m from the floor, reflecting the location of the ears of students sitting in the center of the classroom. The sound source output level was set at 72 dBA which is similar with the "loud" voice of teachers in lower-grade elementary school classrooms considering hearing and concentration abilities of children under the

age of 9 [24]. This level is corresponded with the vocal level when a male talks "loud". The vocal efforts by a regular male are reported in Table 7 [25].

**Table 6.** Finishing materials for the classroom applied in the simulation.

| Classification | | Location | Material | NRC * |
|---|---|---|---|---|
| Common finishing materials | | floor | linoleum tile | 0.015 |
| | | wall | painted on concrete | 0.015 |
| | | window | glass, PVC frame | 0.038, 0.075 |
| | | door | wood | 0.085 |
| | | blackboard | wood | 0.083 |
| | | lecture desk | wood | 0.083 |
| | | desk | wood, metal | 0.083, 0.075 |
| | | chair | wood, metal | 0.083, 0.075 |
| | | cabinet | wood | 0.083 |
| | | ceiling | mineral board | 0.518 |
| $RT_{mid}$ ** | 0.4 s | ceiling | 95% absorption | 0.950 |
| | 0.6 s | | 60% absorption | 0.600 |
| | 0.8 s | | 38% absorption | 0.380 |
| | 1.0 s | | 27% absorption | 0.270 |
| | 1.2 s | | 21% absorption | 0.210 |

* noise reduction coefficient: average of the sound absorption coefficient with 250, 500, 1 k, 2 k Hz. ** $RT_{mid}$: average RT of 500 Hz & 1 kHz.

**Table 7.** Vocal level according vocalization effort of a regular male.

| Vocal Effort | LS, A, 1 m (dB) |
|---|---|
| maximum shout | 90 |
| shout | 84 |
| very loud | 78 |
| loud | 72 |
| raised | 66 |
| normal | 60 |
| relaxed | 54 |

Acoustic factors other than reverberation time are also influenced by the sound absorption rate indoors. As the reverberation time changes, the definition (D50) and speech transmission index (STI) change, and based on these factors, the speech intelligibility performance of a classroom can be estimated. D50 refers to the ratio of direct sound energy to total sound energy up to 50 ms after direct sound reaches a point. It is an acoustic factor indicating the definition of the room. Usually, D50 increases when the reverberation time is shortened, and D50 decreases when the reverberation time is lengthened. The space where speech transmission is mainly performed requires a D50 value of 50~60% [26]. STI is used as an indicator of understanding of speech transmission such as lectures. The values of the STI range from 0 to 1, indicating the degree of speech transmission in the room is "Fair" at 0.45~0.60, "Good" at 0.60~0.75, and "Excellent" at 0.75 or higher [27].

Figure 5 illustrates D50 and STI values according to reverberation time in the virtual sound field created via simulation. It can be seen that the longer the reverberation time is, the lower the D50 and STI values are. Additionally, when the reverberation time ($RT_{mid}$) is between 0.4 s and 1.2 s, D50 ranges from 87% to 46% and STI values from 0.79 to 0.54.

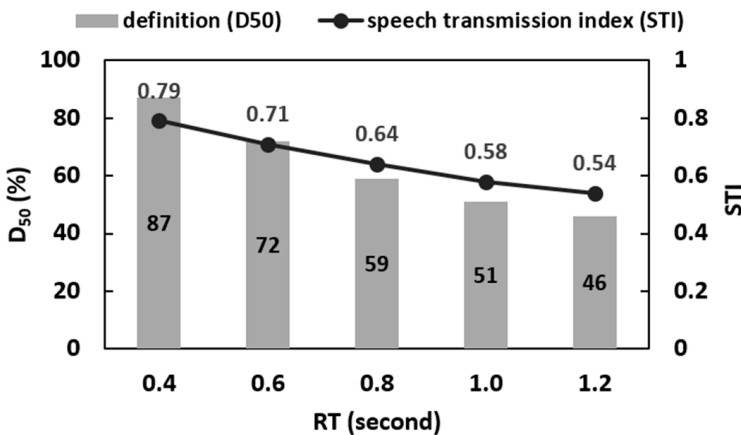

**Figure 5.** Speech intelligibility in the classroom by reverberation time.

### 3.2. Creation of Auralized Sounds

The speech intelligibility tests were conducted using auralization techniques. In five 3D virtual sound fields, binaural impulse responses were recorded by selecting a head-related transfer function (HRTF) and headset model in the computer simulation program and synthesized with the sounds for the speech intelligibility test. The HRTF–headset model was specified as the same as the headset (Sennheiser HD 280, 44,100 Hz) model used to test speech intelligibility. The sounds used in the synthesis were recorded in an anechoic chamber for Korean speech intelligibility tests. Figure 6 illustrates the production process of auralized sounds.

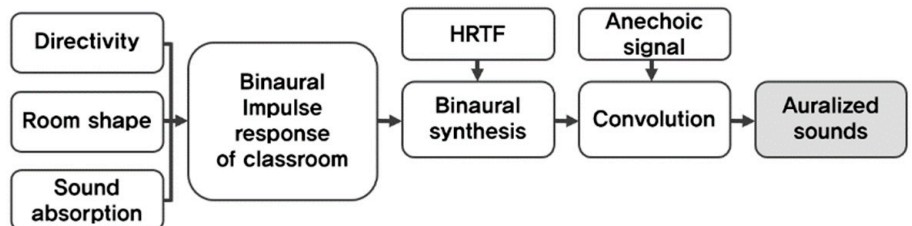

**Figure 6.** Production process of auralized materials.

Generally, a list of nonsense monosyllabic consonant–vowel–consonant (CVC) words [28,29] or a list of phonetically balanced words (PBW) in which phonological phenomena in Korean word pronunciation are evenly distributed is used to evaluate Korean speech intelligibility [30]. In the PBW method, meaningful words of three or more syllables are used. When adults are tested, it is typical to perform the speech intelligibility test by using both the CVC and PBW methods based on a word list consisting of 50 words per set. During the test, a word is auditorily presented, and adult subjects are instructed to write it down within a specified time. Compared to adults, however, the number of words that lower-grade elementary school students can listen to and write down is limited, and their attention spans are shorter. Hence, it is difficult to evaluate children with the same number of words that would be used for an adult's test [31].

Accordingly, the word list used in this study was developed by consulting with three homeroom teachers of lower-grade elementary school students and selecting words appropriate for children. The outcome of the consultation was based on the current curriculum and converged opinions of homeroom teachers of lower-grade students at the elementary schools at which they were working. Opinions of at least 15 teachers were gathered. As a result of the consultation, the PBW method was excluded based on the opinion that accurate speech intelligibility testing would be impossible with three or more syllable PBWs because lower-grade elementary school students have not learned enough such words and cannot understand them. In addition, to the control difficulty

level, monosyllabic CVC words were classified as "easy," "moderate," "difficult," and "impossible" upon consultation, and the final sets of CVC words were created such that children's average test scores would be 75–80 points. Specifically, there were 14 easy, 10 moderate, and 1 difficult CVC words (a total of 25) in a set. A total of five sets of CVC words were developed in this manner, as it was planned to test speech intelligibility in a total of five conditions of reverberation time. To prevent learning during the experiment through the repeated presentation of CVC words, the five sets consisted of entirely different monosyllables. To familiarize the children with the experimental procedure, a practice list consisting of four monosyllabic words was used prior to the main experiment. Table 8 is a Korean-language speech intelligibility test tool.

**Table 8.** Sample list of the speech intelligibility test for children under 10 years of age (Korean).

| practice list | 산[san] | 들[deul] | 별[byeol] | 꽃[kkot] | |
|:---:|:---:|:---:|:---:|:---:|:---:|
| | 호[ho] | 차[cha] | 말[mal] | 워[wo] | 납[nap] |
| | 순[sun] | 칼[kal] | 회[hoe] | 콩[kong] | 철[cheol] |
| evaluation list | 결[gyeol] | 매[mae] | 광[gwang] | 손[son] | 찌[jji] |
| | 쌀[ssal] | 활[hwal] | 듬[deum] | 끼[kki] | 외[oe] |
| | 공[gong] | 의[ui] | 빨[ppal] | 입[ip] | 초[cho] |

The duration of each CVC word and the interval between words were controlled by using Adobe Audition CC program. A word was presented for 1–2 s, and the interval between words was 7 s to allow a sufficiently long time during which the children could hear and write down a word.

### 3.3. Speech Intelligibility Test

Speech intelligibility tests were performed on May 17 and 22, 2021. The total number of subjects was 40: 20 lower-grade elementary school students with incomplete hearing and 20 adults with normal hearing. Hearing-impaired groups are different from the groups with incomplete hearing. Incomplete hearing groups contain the people with no medical hearing difficulties but, they have not yet matured due to young ages. During the tests, classroom doors and windows were closed to minimize the influence of external noises. The tests were performed in identical conditions throughout the experiment, including conditions such as the word list, presentation order, and output level. Detailed information on the number and age of subjects, test dates, and background noise levels is presented in Table 9.

**Table 9.** General information on speech intelligibility test.

| | **Children** | **Adults** |
|:---:|:---:|:---:|
| No. of participants | 20 | 20 |
| Age (number of persons) | 9(10), 10(10) | 24.4 * |
| Background noise | 24.4 dBA | 23.8 dBA |
| Temperature | 23.2 °C | 21.3 °C |
| Humidity | 40.2% | 35.7% |

* average age.

Subjects were instructed to listen to each test word through a headphone (Sennheiser HD 280) and write it down on the answer sheet. How accurately they perceived the words was evaluated by computing the proportion of correct answers. The headphone's output level was set to 64 dBA to reflect the sound level in the center of a classroom when the teacher's voice level is 72 dBA. To ensure that the sound output levels would be identical in all headphones, a sound level meter was used to measure a test sound played through the headphones, and the level was adjusted. A diagram of the outputting of the sounds used to test speech intelligibility is presented in Figure 7.

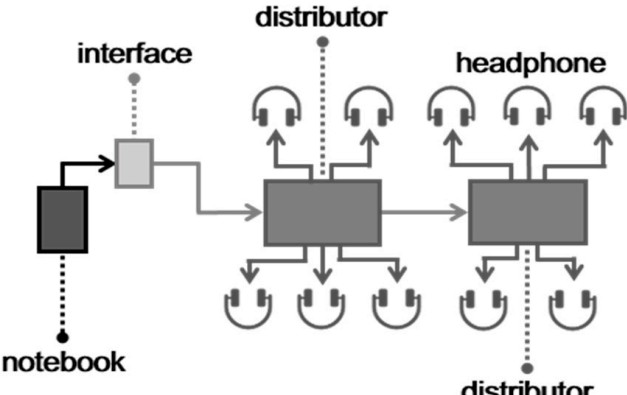

**Figure 7.** Equipment configuration diagram.

To prevent familiarization with reverberation, the auralized sounds were provided randomly, as shown in Table 10. To account for hearing fatigue, subjects took a 2–3 min break between sets.

**Table 10.** Order of speech intelligibility test.

| Test Order | RT (s) |
|:----------:|:------:|
| 1 | 1.0 |
| 2 | 0.4 |
| 3 | 0.8 |
| 4 | 0.6 |
| 5 | 1.2 |

## 4. Research Result

### 4.1. Speech Intelligibility Test Results

Speech intelligibility tests were accomplished to 20 lower-grade elementary school students and 20 adults with five reverberation conditions. When scoring the speech intelligibility test answers, 4 points were given to each question so that the total score was 100 points. Homophones were treated as correct answers. For example, "갇[gt]" and "같[gt]" were treated as the same answer because their pronunciations are the same, [gt]. In addition, because it is difficult for lower-grade elementary school students to differentiate the sounds of "consonant + ㅔ[e]" and "consonant + ㅐ[ae]" an answer with a vowel of either "ㅔ[e]" or "ㅐ[ae]" was scored as correct if the consonant was correct. For instance, "제[je]" and "재[jae]" were treated as the same answer. The average speech intelligibility test scores of children and adults at each reverberation time are shown in Table 11.

**Table 11.** The results of speech intelligibility testing (average scores).

| RT (s) | | 0.4 | 0.6 | 0.8 | 1.0 | 1.2 |
|:------:|:--:|:---:|:---:|:---:|:---:|:---:|
| Score | children * | 74.8 | 79.4 | 72.2 | 72.0 | 68.4 |
| | adults ** | 85.4 | 87.8 | 85.4 | 79.8 | 78.8 |

* under 10 years of age ** 20s (normal hearing).

The children's average score was highest as 79.4 when the reverberation time was 0.6 s. At all other values, they scored 75 or lower, and thus, the difference from the highest score ranged from 4.6 to 11. In adults with normal hearing, the average speech intelligibility test score was highest as 87.8 when the reverberation time was 0.6 s.

Figure 8 shows the average speech intelligibility test scores of children and adults for each reverberation time. Figure 9 illustrates the rates of increase and decrease in speech intelligibility test scores.

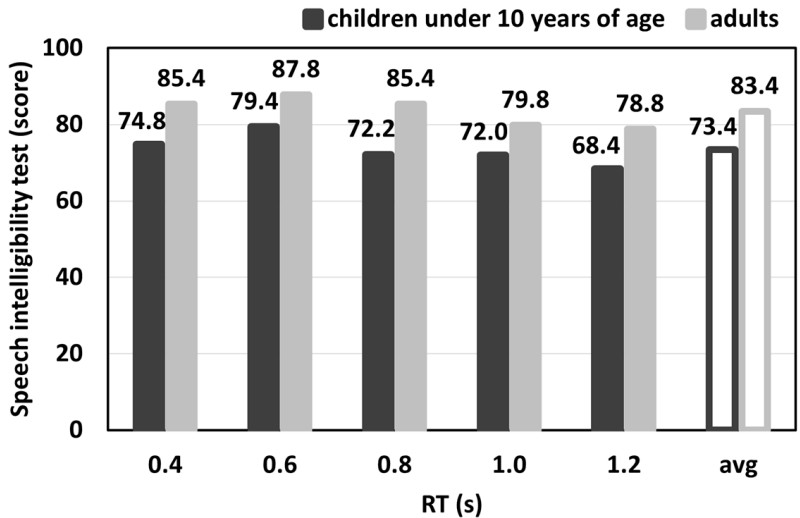

**Figure 8.** Comparison of speech intelligibility test scores in children and adults.

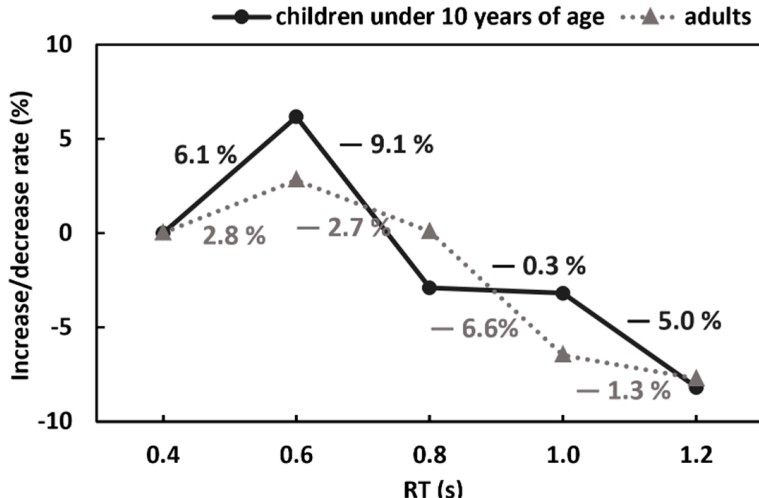

**Figure 9.** The rate of increase and decrease in the speech intelligibility test scores.

At all the conditions of reverberation time, average scores of lower-grade elementary school students are less than the scores of adults by approximately 10 points. It is speculated that children's score was low because the speech perception ability of lower grade elementary school students was significantly lower than that of adults. Additionally, in both children and adults, the longer the reverberation times, the lower the scores.

In children, the score was the highest, 79.4, when the reverberation time was 0.6 s, and the rate of decrease was the greatest, 9.1%, when the reverberation time increased from 0.6 s to 0.8 s. Based on this finding, it is believed that a reverberation time of 0.6 s is the most appropriate for lower-grade elementary school classrooms. It is clear that the standard for reverberation time in lower-grade elementary school classrooms should be shorter than the standard for middle and high school classrooms, which is 0.8 s.

On the other hand, adults scored 85 points or higher when the reverberation time was between 0.4 s and 0.8 s, and under 80 points when it was 1.0 s or longer. The rate of decrease was the greatest, 6.6%, when the reverberation time was between 0.8 s and 1.0 s. Considering that students aged 14 or older show the same level of speech perception ability as adults, the current reverberation time standard for middle and high school classrooms in Korea, 0.8 s, seems appropriate.

### 4.2. Reverberation Time Appropriate for Lower-Grade Elementary School Classrooms in Korea

A standard for appropriate speech transmission in lower-grade elementary school classrooms in Korea was derived based on the speech intelligibility tests performed with various conditions of reverberation time. The proposed standard for the reverberation time is below 0.6 s.

Even in the same listening environment, speech intelligibility varies depending on the language, and the auditory cognition ability differs at different ages. Therefore, the standard derived herein should be applied only in lower-grade elementary school classrooms in Korea. The derived reverberation time standard for lower-grade elementary school classrooms is reported in Table 12.

**Table 12.** Proposed reverberation time standard for Korean lower-grade elementary school classrooms.

| Volume | Reverberation Time ($RT_{mid}$) * |
|---|---|
| below 185 $m^3$ | below 0.6 s |

* average RT of 500 Hz & 1 kHz.

However, this study does not recommend 0.4 s of RT because excessively low RT causes a loss of sound level in the speech space [25]. The speech intelligibility test result also showed that the score at 0.4 s of RT was lower than that at 0.6 s. Therefore, it is recommended to design the lower-grade classrooms of Korean elementary schools in the range of 0.6 s.

The reverberation time standard suggested in this study can be applied only to the situation when teachers speak to the children from the front. Currently, the Korean teaching method is generally conducted by teachers looking at students from the teacher's desk. Also, this standard cannot be applied to special classrooms for learning music, art, and physical education.

Table 13 compares the reverberation time standards of the US, UK, and Korean middle and high school classrooms with the present study. The standard for middle and high school classrooms in Korea is 0.8 s, which is longer than the standard for lower-grade elementary school classrooms derived in this study (0.6 s) by approximately 0.2 s. Thus, lower-grade elementary school classrooms require a space with higher sound clarity compared to middle and high school classrooms. The reverberation time standards in the United States are provided according to the volume of the classroom regardless of age, and the standard is 0.6 s or less. Meanwhile the United Kingdom provided the standard without limiting the volume of the classroom. The standard has limits regarding age, and the reverberation time standard in elementary school is 0.6 s or less.

**Table 13.** Comparison of reverberation time standards in various countries.

| Country | Grade | Volume Limit | RT |
|---|---|---|---|
| Korea | lower-grade elementary school (example) | below 185 $m^3$ | below 0.6 s |
| | middle and high school * | below 220 $m^3$ | below 0.8 s |
| US ** | all school | below 283 $m^3$ | below 0.6 s |
| UK *** | elementary school | no limitation | below 0.6 s |
| | middle and high school | no limitation | below 0.8 s |

* Park, C.J.; Haan, C.H. Initial study on the reverberation time standard for Korean middle and high school classrooms using speech intelligibility tests. Buildings 2021, 11, 354. ** ASA S12.60: acoustic performance standard, design requirements, and guidelines for schools. *** Building Bulletin 92: Acoustic Design of Schools a Design Guide.

These standards are related to the characteristics of the language being used, the methods of education, and the construction type in the country. Therefore, standards for classroom acoustic environments should be developed for the country where the standards will be applied. In addition, the auditory cognition ability corresponding to student age should be factored into the classroom acoustics standards.

### 5. Conclusions

The aim of this study was to recommend a reverberation time appropriate for lower-grade elementary school classrooms. To do so, a standard classroom of a lower-grade elementary school was reproduced, and auralized sounds with various reverberation times were created using computer simulation. Using the auralized sounds, speech intelligibility tests were administered to lower-grade elementary school students and adults, and the test scores were analyzed. The conclusions can be summarized as follows:

(1)　In both lower-grade elementary school students and adults, speech intelligibility test scores decreased as the reverberation time increased.

(2)　The average scores of the students were lower than those of adults. Thus, it can be inferred that speech perception ability is lower in children than in adults.

(3)　For lower-grade elementary school students, the percentage of correct answers fell by 9.1%, when the reverberation time was increased from 0.6 s to 0.8 s. On the other hand, adults' percentage of correct answers fell by 6.6%, when the reverberation time was increased from 0.8 s to 1.0 s.

(4)　It is believed that an educational space with a reverberation time of 0.6 s or less is the most appropriate for lower-grade elementary school students.

This study was conducted with only 20 subjects per group and hence, it is difficult to generalize the findings. In addition, the effects of facial masks and screens were not considered. According to a recent paper, acoustic transmission is influenced by protections such as face masks [32]. Moreover, not only the reverberation time but also the background noise is an important factor affecting classroom acoustics, and it should also be regulated. In the future, additional research should be performed with lower-grade elementary school students to recommend acoustics standards for reverberation time and background noise appropriate for the students.

Based on the findings, classroom design guidelines for interior finishing materials and layout can be developed to implement acoustic standards for lower-grade elementary school classrooms in educational spaces and ultimately to provide an educational environment appropriate for the hearing of lower-grade elementary school students.

**Author Contributions:** Conceptualization, C.-H.H.; methodology, A.-H.J., C.-J.P. and C.-H.H.; software, A.-H.J.; validation, A.-H.J., C.-J.P. and C.-H.H.; formal analysis, A.-H.J. and C.-H.H.; investigation, A.-H.J., C.-J.P. and C.-H.H.; resources, C.-H.H.; data curation, A.-H.J.; writing—original draft preparation, A.-H.J.; writing—review and editing, C.-H.H.; visualization, A.-H.J.; supervision, C.-H.H.; project administration, C.-H.H.; funding acquisition, C.-H.H. All authors have read and agreed to the published version of the manuscript.

**Funding:** This work was supported by the National Research Foundation of Korea (NRF), grant funded by the Korean government (MSIT) (NRF-2020R1A2C2009963).

**Informed Consent Statement:** Informed consent was obtained from all subjects involved in the study.

**Conflicts of Interest:** The authors declare no conflict of interest.

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
