# Peer review of "Investigation of the Appropriate Reverberation Time for Lower-Grade Elementary School Classrooms Using Speech Intelligibility Tests"

_buildings, doi:10.3390/buildings12060808_

Round 1
Reviewer 1 Report
- Abstract – in the last line “low” should be “lower”.
- Why is it not explicitly stated why research on classroom acoustics (which has already been conducted extensively internationally) has to be repeated for Korea. Is there something about the Korean language that potentially would make the RT requirement different from in English speaking countries? Ref 13 seems to be given as a justification but this is not explained. Why is the required information not available from the research detailed in ref 12?
- Line 63: ‘test was’ should be ‘tests were’
- Line 70: Could be better expressed as “The age range for children from elementary to high school is 6- 18”
- Table 4: Might be helpful to explain the “Classification” and “Grade of classroom”
- Line 114: It would be usual to provide a reference for standards that are mentioned
- Line 116: What was the directionality of the loudspeaker? Did it match that of a human speaker?
- Fig 2 labelling: This could be rearranged to be clearer for the reader. It would be better to omit the word “area” from the legend.
- Line 138 (and following): The frequency dependence of the RT should be acknowledged and discussed. In the text RT is used as though it has a single value and its frequency dependence is not significant.
- Line 55: This illustrates the above point. It says “a reverberation time” as though a single value is all that is needed.
- Line 164 section 3.1: Is the architecture of the Korean classrooms always a rectangle with a flat ceiling? How might the ceiling design (the place for the room absorption) affect the SNR and integrated reflections?
- Line 182” What is meant by “inter-aural level difference”? In most literature it means the level difference between the two ears of a listener. Is that the case here? If so why is it large?
If it is suggesting that for listeners sitting near a wall one ear is more strongly affected by the strength of the wall reflection than the other this should be explained and the significance of this difference for those listeners spelled out.
- Line 189 and following: How was the background noise in the measured rooms taken into account in the presentation system? Did this vary with the RT?
- Line 202: A reference is required for the relationship between RT and D50 and STI.
- Lines 230-234: I suggest this is rewritten. It is not clear what is meant by “converged”.
- Line 258: The phrase “incomplete hearing” is ambiguous. I suspect it means not fully developed to adult maturity but normal for their age, but it could also mean hearing impaired. This should be clarified.
- Line 277: It isn’t clear how “auditory learning of reverberation time” would affect the results.
- Lines 284-289: It should be made clear whether the scoring is of whole words or whether the phonemes are scored individually i.e. does a CVC word have a potential score of 1 or 3.
- Line 281 and following: In this section I feel there should be discussion of what change in the %age in the speech intelligibility score might be considered as significant. For example, an RT of 0.6 could be considered an optimum value if the change in %age from 0.4 to 0.6 is considered as significant and repeatable. Also there should be some discussion of why it might be that as the RT gets lower than 0.6 the intelligibility score is decreasing. Is this found in comparable studies elsewhere and considered to be a real effect in rooms or is it an artefact of the presentation system?
This is an important point as Table 13 shows that the criteria for other countries is typically “below” a certain value suggesting that the lower the RT the better.
- Line 327: The legend for the table should be amended to read “Proposed Reverberation Time standard ….”
- Finally, it should be emphasised that this study only applies to the style of teaching where the teacher addresses the children from the front (sometimes referred to as ‘sage on the stage’). Some comment should be made about the relevance to Korea of the style of teaching used in so-called ‘modern learning environments’ where most of the learning is in small groups and where there is significant student-generated noise.
Author Response
Dear Reviewer,
We appreciate for your valuable comments on our manuscript.
Responses to comments can be found in the attachment.
Please see the attachment.
Authors

Reviewer 2 Report
This is an iteresting paper, thanks to the authors.
Some remarks and suggestens are given below:
Line 114: It is written: "... KS F 2864 ..." The inexperienced reader does not know what this means. Additional this reference is not cited.
Line 176: It is written: "... sound simulation software, Odeon." It is suggested to write: ... using an acoustics simulation software [x]. A reference should be provided in the literature.
Line 184: It is wirtten: "... materials.[12]" The fullstop shoud be after the bracket, i.e. materials [12].
Line 196: same as in Line 184
Line 198: same as in Line 184
Line 207: There are some numbers given, describing a decrease. However, it is not explained what the number means. It is suggested to provide some explanation, that a smal number of e.g. STI is good and that the STI varies between 0 and 1. This would help to understand the numbers.
It is not explained what D50 and STI stand for. An explanation is necessary to understand the text.
It should be stated that in classrooms a STI > 0.6 is good. This is documented e.g. in the standard IEC 60268-16:2020 - Sound system equipment - Part 16: Objective rating of speech intelligibility by speech transmission index. This reference is missing in the paper.
Line 250, the number is in the middle of the sheet, should be on the right margine.
Tabel 8: The text in this table is in Korean. A translation should be used in order to understand the meaning of the table.
Line 339: It is written: "... no restriction in classroom volume." If a reverberation time is given as a limit, it is not needed to restrict the room volume. A room volume limit makes sense for the use of space but not for the acoustic suitability, since the reverberation time is limited. This is true due to the fact that room volume and reverberation time are mathematically linked.
A general remark:
The German Standard DIN 18041 Acoustic quality in rooms — Specifications and instructions for the room acoustic design, states:
The reverberation time requirements for good acoustic quality depend on room volume and the usage type of the room. For rooms of group A, the following usage types are differentiated: ...
- A3: “Education/Communication” plus “Speech/Presentation inclusive”;
- A4: “Education/Communication inclusive”;
Furthermore DIN 18041 states:
By people with impaired hearing, shorter reverberation times are perceived as being acoustically more favourable for spoken communication. The same applies for communication in a foreign language and for communication with people who for other reasons require increased speech intelligibility, e.g. people with speech or speech processing problems, concentration or attention problems, or general capacity impairments. In cases of doubt, in rooms for voice information and communication, the reverberation times should rather be shorter than longer.
Author Response

(The authors gave the same response as above.)

Reviewer 3 Report
This is a strong research report with an educationally meaningful research premise (from abstract): "Since speech recognition performance is significantly low at the age of nine or younger, acoustic performance standards of classrooms for young children should be investigated" supported by (from intro): "To provide students with an acoustic environment adequate for learning, classroom acoustics standards should be available." The report's conclusion of an RT recommendation time is well-supported by the systematic study details, and explained with convincing data. However, there are two questions (at the end, below) that can be addressed for clarification, and one that is a problematic conclusion that requires further explanation.
Overall, the paper is well-written and includes extensive detail:
Background information supports the research design, such as (lines 56-59) "Additionally, in classrooms with a similar acoustic condition, factors affecting speech intelligibility differed depending on whether the language was Korean, English, or Chinese.[13] Therefore, classroom acoustics standards should be developed specifically for the language used in the classroom." We also learn that in Korea, permitted background noise levels in classrooms are higher than in comparison countries, and that (line 77) "Researchers in countries other than Korea have also recognized the need to differentiate classroom acoustics standards according to age." These and other background factors substantiate the study design.
Ecological validity for the teaching and learning context is accounted for in the classroom acoustical measurements (line 118-119): "The sound source was positioned in the center of the podium in front of the 118 blackboard in consideration of the teachers’ location, at a height of 1.5 m from the floor in consideration of the average location of the mouth of an adult" and (lines 124-125): "Sound receivers (microphones) were located at 1 m from the 124 floor in consideration of the location of the ears of lower grade elementary school students 125 sitting on a chair." Careful attention to the context of receiver positions near windows or room center, and factors such as construction time/materials are also noted. In addition, the experiment stimuli was carefully developed for real-world applicability considering the age of the students concerned (lines 230-231).
The figures are excellent visualization tools to elaborate the data collection and modeling research. The tables provide important data and enable a quick comparison of parameters.
Two Questions for Clarification:
1) (lines 211-214): "In five 3D virtual sound fields, binaural impulse responses were recorded by selecting head-related transfer function (HRTF) and headset model in the computer simulation program and synthesized with the sounds used in the speech intelligibility test."
What were the criteria used to select the HRTF-headset model?
2) The only problem in the study appears to be the limitation of 0.6s RT in the conclusion. Despite the fact that study participants performed better at 0.4s RT, the recommendation is 0.6s. That decision is not explained. Reviewing Table 13, it is shown that RTs in the US and UK for all school levels should be "below 0.6s". Given that lower RTs appear to be bette, why did this study limit the RT to 0.6 seconds, rather than seeking the optimal level below 0.6s? That point should be explained and justified -- and perhaps a followup study to find the lower reasonable limit is warranted.
Author Response

(The authors gave the same response as above.)

Round 2
Reviewer 1 Report
Thank you for the revision.